# Dual Responsive Dependent Background Color Based on Thermochromic 1D Photonic Crystal Multilayer Films

**DOI:** 10.3390/polym14235330

**Published:** 2022-12-06

**Authors:** Yejin Kim, Seo Hyun Kim, Henok Getachew Girma, Seungju Jeon, Bogyu Lim, Seo-Hyun Jung

**Affiliations:** Center for Advanced Specialty Chemicals, Korea Research Institute of Chemical Technology (KRICT), Ulsan 44412, Republic of Korea

**Keywords:** thermochromic, 1D photonic crystal, dual responsive, humidity, temperature responsive

## Abstract

In this paper, we present dual responsive one-dimensional (1D) photonic crystal (PC) multilayer films that utilize a high-humidity environment and temperature. Dual responsive 1D PC multilayer films are fabricated on precoated thermochromic film by sequential alternate layer deposition of photo-crosslinkable poly(2-vinylnaphthalene-co-benzophenone acrylate) (P(2VN-co-BPA)) as a high refractive index polymer, and poly(4-vinylpyrollidone-co-benzophenone acrylate) P(4VP-co-BPA) as a low refractive index polymer. The thermochromic film shows a vivid color transition from black to white at 28 °C. Three different colors of thermochromic 1D PC multilayer films are prepared by thickness modulation of P(4VP-co-BPA) layers, and the films on a black background exhibit visible spectrum color only in a high-humidity environment (over 90% relative humidity (RH)). For the three films placed on a hands display, three different composite colors are synthesized by the reflection of light, including yellow, magenta, and cyan, due to the changing of backgrounds from black to white with temperature. Additionally, the films show remarkable color transitions with reliable reversibility. The films can be applied as anti-counterfeiting labels and can be used for smart decoration films. To the best of our knowledge, this is the first report of dual response colorimetric films that change color in various ways depending on temperature and humidity changes, and we believe that it can be applied to various applications.

## 1. Introduction

The rapid rise in counterfeit products is raising interest in anti-counterfeiting technology. Consequently, various anti-counterfeiting technologies, such as radio frequency identification (RFID), quick response (QR) codes, barcodes, holograms, and color-changing technologies, have been developed. Photonic crystals (PCs), one of the color-changing technologies, are attracting attention owing to their color tunability, fast response time, low cost, and remarkable reversibility and repeatability. In PCs, dielectric materials with different refractive indices are periodically arranged. Therefore, PCs selectively reflect light of a specific wavelength and display light composite colors depending on the background color. Recently, responsive photonic crystals (RPCs) have been extensively investigated for real-time monitoring colorimetric sensors. Moreover, RPCS have gained attention as materials with photonic bandgap properties that can be tuned by different external chemical and physical stimuli, such as humidity [1,2,3,4,5,6,7], light [8,9], temperature [10,11,12], pH [13,14], pressure [15,16], solvents [17,18,19], electricity [20,21], and biomolecules [22,23]. Additionally, RPCs have been investigated for their unique optical properties in devices such as sensors, displays, lasers, anti-counterfeiting devices, and smart decoration films.

By integrating pigments into photonic crystal, we assume the integration of both structural and pigmentary color due to diversification of colors. The most common approach is to integrate thermal- or light-responsive chromatic pigments with photonic crystals. Huang et al. introduced red and yellow thermochromic pigments and viologen-based electrochromic materials into a photonic crystal film [24]. This film exhibited diversified color through coexisting interactive structural and pigmentary components by external stimuli. In addition, Liu et al. reported a thermal-responsive photonic crystal (TSPC)-based fluoran dye (heat-sensitive red TF-R2) into SnO_2_ inverse opals [25]. Furthermore, TRPC displayed pigmentary color below the phase transition temperature of fluoran dye and exhibited structural color above it. Li et al. reported thermochromic photonic crystal (TCPC) films based on a mixture of PS@PEA@P2EHA CIS colloidal particles with heat-sensitive pigment microcapsules using the bending-induced ordering technique (BIOT) [26]. Furthermore, TCPC films exhibited switchable colors at different angles and temperatures with remarkable reversibility and higher color visibility. The integration of chromatic pigments with photonic crystals can be used for anti-counterfeiting, information encryption, decorations, etc.

In this study, we demonstrate a thermal and high-humidity environment responsive 1D PC multilayer film based on a colorimetric system. Compared to three-dimensional (3D) and two-dimensional (2D) PCs, one-dimensional (1D) PCs have several advantages, including a simple structure, good optical properties, a simple manufacturing process, and low cost [27]. Thermochromic film that changed from black to white color was first fabricated on a transparent polyethylene terephthalate (PET) substrate by bar coating. The film displayed a vivid color transition from black to white at 28 °C. The high refractive index polymer poly(2-vinylnaphthalene-co-benzophenone acrylate) (P(2VN-co-BPA)) and the low refractive index polymer poly(4-vinylpyrollidone-co-benzophenone acrylate) (P(4VP-co-BPA)) were alternately pre-coated on the thermochromic film to fabricate the high-humidity responsive 1D PC multilayer film. Red, green, and blue (RGB) colors of 1D PC multilayer films were prepared by thickness modulation of P(4VP-co-BPA) layers. With a black background, the film exhibited a visible spectrum color change under a high-humidity environment (over 90% RH). In contrast, thermochromic 1D PC multilayer films changed color from black to white with temperature, and the films displayed three different composite colors, including yellow, magenta, and cyan. The films combining the 1D PC and the thermochromic film could have significant potential in applications such as anti-counterfeiting or smart decoration films.

## 2. Experimental

### 2.1. Materials

2,2-Azobisisobutyronitrile (AIBN; 98%, JUNSEI, Tokyo, Japan) and acryloyl chloride (96%, Merck KgaA, Rahway, NJ, USA) were used after recrystallization with methanol and purification through a basic alumina-packed column, respectively. 2-Vinylnaphthalene (95%, Sigma-Aldrich, St. Louis, MO, USA), triethylamine (TEA, 99%, Tokyo Chemical Industry Co., Ltd.—TCI, Tokyo, Japan), 4-hydroxybenzophenone (98%, Alfar Aesar, Ward Hill, MA, USA), N,N-dimethylformamide (DMF, 99.9%, SAMCHUN, Seoul, Republic of Korea), 1,4-dioxane (99%, Sigma-Aldrich, St. Louis, MA, USA), n-propyl alcohol (99.5% DAEJUNG, Siheung, Republic of Korea), acetone (99.5% DAEJUNG), toluene (99.5% DAEJUNG), thermochromic powder (Camel Chemicals, Busan, Republic of Korea), polyurethane resin (Chokwang Paint, Busan, Republic of Korea), and thinner (Chokwang Paint, Busan, Republic of Korea) were used as received.

### 2.2. Instrumentation

^1^H NMR spectroscopy was performed on an Avance III HD 300 MHz spectrometer (Brucker, Billerica, MA, USA). The molecular weights were determined by gel permeation chromatography (GPC) (1260 Infinity, Agilent, Santa Clara, CA, USA) using polystyrene standards with DMF eluent at a flow rate of 1 mL/min at 30 °C. Spectral reflectance was measured using a UV–vis spectrometer (USB4000, OceanOptics Inc., Dunedin, FL, USA). The refractive index of each polymer was observed by ellipsometry (HORIBA Scientific, Kyoto, Japan, UVISEL).

## 3. Synthesis

P(2VN-co-BPA) was synthesized as mentioned previously [28].

### 3.1. P(4VP-co-BPA) Preparation

4-Benzophenone acrylate (5 g, 1.98 mmol), N-vinylpyrrolidone (25 mL, 23.4 mmol), AIBN (0.1 g, 0.06 mmol), and 15 mL of 1,4-dioxane were added to a Schlenk flask, and the flask was deoxygenated by three freeze-pump-thaw cycles and nitrogen purging for 20 min. Polymerization started at 70 °C in an oil bath and was carried out overnight. Subsequently, the polymerization was quenched by exposing the solution to air. The synthesized polymer was precipitated twice in diethyl ether and filtered, and the product was dried under vacuum overnight at 80 °C. ^1^H NMR (300 MHz, DMSO-d6) δ 7.58 (9H, m), 3.14 (2H, s, N-CH2-CH2), 2.06 (2H, s, CH2-CH2-(C=O)), 1.87 (2H, s, CH2-CH2-CH2).

### 3.2. Fabrication of the Thermochromic Film

The coating mixtures consisting of 10 wt% thermochromic pigment, 10 g of thermochromic powder (Camel Chemicals), 45 g of polyurethane resin (Chokwang Paint), and 45 g of thinner (Chokwang Paint) were stirred vigorously for 30 min. The mixture was deposited on a PET film substrate using a wire-bar coater at a speed of 25 mm s^−1^. The film was then dried in air at room temperature for 1 h. The thickness of the thermochromic layer was 15 μm, as measured by a surface profiler.

### 3.3. Fabrication of 1D PC Films

Three films were prepared as follows: 1.6, 2.0, and 2.6 wt% P(4VP-co-BPA), respectively, was dissolved in 1-propanol and acetone (proportion 2:1), and 1.3 wt% P(2VN-co-BPA) was dissolved in toluene and was passed through a 0.45 µm PTFE syringe filter before spin-coating. Additionally, the two polymer solutions were alternatively spin-coated onto a PET film at 2500 rpm for 12 s. When each spin-coated layer was applied, the film was exposed under a UVA lamp (652 mJ/cm^2^). This process was repeated to generate 10 layers. The production conditions and thickness of each film are summarized in Table 1.

## 4. Results and Discussion

To fabricate thermal and high-humidity responsive 1D PC multilayers, first, we synthesized P(2VN-co-BPA) (Mn = 51,000 g/mol, Mw/Mn = 2.84) and P(4VP-co-BPA) (Mn = 30,400 g/mol, Mw/Mn = 2.33) as high (HP) and low (LP) refractive index polymers via free radical polymerization. It was confirmed that the synthesized polymers were synthesized through ^1^H NMR analysis, and these spectra are shown in Appendix A of the Supporting Information. The GPC results of each polymer are shown in Appendix A of the Supporting Information. Additionally, BPA contains 5–10 mol% to achieve sufficient crosslinking density for film stability, as shown in Figure 1a. Before fabricating 1D PC multilayers, we investigated HP and LP monolayers’ refractive index and thickness by ellipsometry. Furthermore, HP and LP samples were prepared by 2.5 wt% of P(2VN-co-BPA) in toluene, and 1.6, 2.0 and 2.6 wt% of P(4VP-co-BPA) in propanol and acetone mixture solutions were spin-coated onto silicon wafer at 2500 rpm for 12 s. The refractive indices were 1.64 for P(2VN-co-BPA) and 1.51 for P(4VP-co-BPA), and the monolayer thickness of 2.5 wt% of P(2VN-co-BPA), and 1.6, 2.0, and 2.6 wt% of P(4VP-co-BPA) were 50, 90, 130, and 165 nm, respectively (Figure 1b,c).

In order to realize a dual responsive film that changes color depending on temperature as well as humidity, films based on thermochromic dye were intended to be used. To induce a color transition using the temperature of a human hand, thermochromic dye with a color transition at 28 °C was selected with black and white color because 1D PC multilayer films display two different colors depending on the background. A black background exhibited the full visible spectrum color, and a white background displayed three different composite colors synthesized by the reflection of light, including yellow, magenta, and cyan [28].

Thermochromic film was prepared by bar coating on a transparent PET substrate. Before fabricating 1D PC multilayers, we investigated the color transition of thermochromic film with temperature. Figure 2b shows the color transition of thermochromic films from black to white at a temperature of >28 °C, and the light value of L* represents the dark L* = 0 and the white L* = 100. From the RGB value, L* of this film was calculated. For a temperature < 27 °C, the L* value was from 14 to 27, and for a temperature of >27 °C, it was from 83 to 86. The produced thermochromic film changed from black to white very quickly with human body temperature only, and it was confirmed that it was suitable for use for the purpose of this study.

As shown in Figure 2a, the 1D PC multilayer films were prepared by sequentially depositing alternating layers of P(2VN-co-BPA) and P(4VP-co-BPA) on a pre-coated thermochromic film by spin-coating each layer, followed by chemical immobilization of the surface by irradiation with 652 mJ/cm^2^ UV light. The distinct initial colors of the 1D PC multilayer films can be controlled by varying the thickness of the high- and low-refractive-index polymers, which can be controlled by changing the solvent type, the concentration of the copolymer in the solution, and the spin-coating speed. As shown in Figure 2c, three different initial color films are prepared, i.e., purple, green, and red, by modulating the P(4VP-co-BPA) layer thickness through the control solution concentration. This indicates that the photonic stop bands (PSBs) were adequately controlled through the thickness change.

The reflectance spectra of three colors films show 425 nm for purple, 497 nm for green, and 644 nm for the red film (Figure 2d). The theoretical PSB was calculated by the measured ellipsometry refractive index and thickness of HP and LP. Additionally, it was compared with the theoretical PSB, which was in good agreement with the experimental results. These results confirm that the HP and LP layers are well formed with uniform thicknesses.

If properly designed with the thickness control of each refractive index layer of 1D PC film, the swelling of the LP layer increases with humidity, leading to a redshift and corresponding color shift of the PSB. Three different 1D PC films with a black background were placed in a sealed digital temperature and humidity chamber, exposed to different RH levels, and their response to high-humidity environments was investigated by changing colorimetric UV–Vis reflectance spectra with varying humidity. Three 1D PC films show color differentiation maps with changing RH from 30 to 90%, as shown in Figure 3a. When the RH increased from 30 to 90%, purple, green, and red films display a vivid color change from purple to cyan, green to orange, and red to colorless, respectively. The purple film shows little color change, whereas the green and red 1D PC films show a relatively large color change. These results can also be confirmed through reflectance measurements. The reflectance spectra of the three films are displayed in Figure 3b. The PSB of three films showed a red shift by 72, 105, and 139 nm, respectively. This is because the thickness of the LP layer in the purple film is relatively thin and the expansion by humidity is small. Conversely, the LP layer of the red film is the thickest and expands greatly with humidity, so the change in PSB is large.

For application as a dual responsive anti-counterfeiting film, we aimed to fabricate a film that could identify a counterfeit by human breath and body temperature, as depicted in Figure 4a. A film that changes according to humidity and temperature can be a powerful but very simple anti-counterfeiting technology, because it can implement color changes only with the person’s breath and body temperature without the use of a special device. This requires the film to exhibit color transitions only in very humid environments created by exhalation saturated with water vapor. For the three 1D PC films in Figure 4b, the color transition was monitored over time after each sample was exhaled. The color differentiation map of the three films exhibited a distinct color transition from purple to cyan, green to orange, and red to colorless. In addition, the three films displayed a color transition within 10 s, and the original color was recovered within 20 s.

As shown in Figure 4c, we investigated the swelling dynamics of chameleon films by measuring their time-dependent changes with a UV–vis spectrometer after a human blew on them. The initial PSBs of the three 1D PC films were observed at 470 nm for purple, 500 nm for green, and 642 nm for the red film, and when people blew on the PC surface, water vapor diffused into the film instantaneously. The PSB at the final peak position was close to 552 nm for purple, 641 nm for green, and 782 for red film after 13 s of blow spraying, and as the water vapor disappeared, the PSB of the 1D PC changed again and the film color recovered. Since the thermochromic film can change color from black to white only with human body temperature, and the 1D PC film can quickly change color only with human breath, we have demonstrated the possibility that the dual response color conversion film can be implemented well.

Figure 4d shows three films with different colors on the black and white backgrounds with temperature. The three 1D PC films on a black background reflected purple, green, and red colors, and on a white background they reflected yellow, magenta, and cyan colors. This is because the same 1D PC film looks different depending on the background color. Through these results, we successfully implemented a dual response color conversion film that can change color depending on humidity and temperature.

The stability of the film under repeated humidity sensing is crucial for practical applications. As shown in Figure 5, the correct movement of PSBs was observed for three films under repeated blowing by humans over a period of 10 cycles, and the films displayed excellent durability. The films display vivid color transitions with reflections of colors and complex colors. Additionally, the dual color change films have significant potential as anti-counterfeiting films.

## 5. Conclusions

In this study, dual responsive films were fabricated, utilizing a high-humidity environment (over 90%) and temperature against a background color 1D PC multilayer film. The thermochromic 1D PC films were fabricated by the sequential deposition of alternating layers of P(2VN-co-BPA) and P(4VP-co-BPA) on thermochromic films via bar- and spin-coating, followed by surface immobilization with UV irradiation. The thermochromic films showed vivid color transitions from black to white at 28 °C. Additionally, 1D PC multilayer films exhibited distinct RGB reflection color and color transition only with high RH. In addition, the films exhibited a vivid visible color transition via human blowing on a black background, and three different composite colors in response to human body temperature on a white background, including yellow, magenta, and cyan. The films can be applied as an anti-counterfeiting label and can be used as smart decoration film.

## Figures and Tables

**Figure 1 polymers-14-05330-f001:**
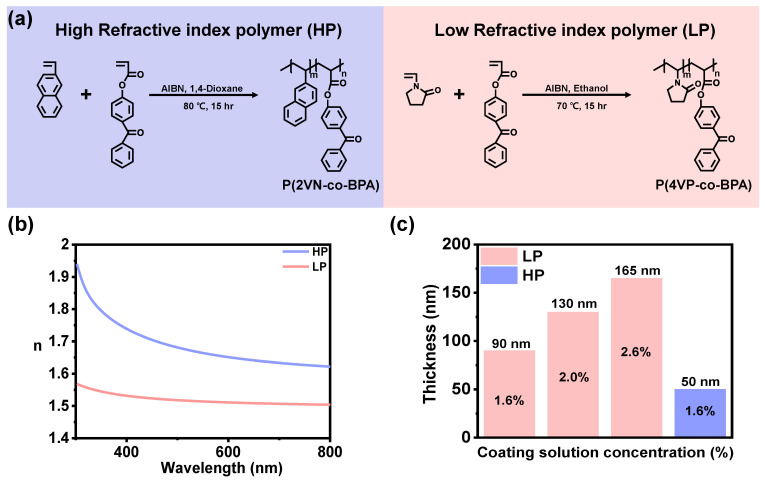
(**a**) Synthesis process of photo-crosslinkable P(2VN-co-BPA) and P(4VP-co-BPA) as high/low refractive index polymers. (**b**,**c**) Monolayer refractive index and thickness of HP and LP polymers.

**Figure 2 polymers-14-05330-f002:**
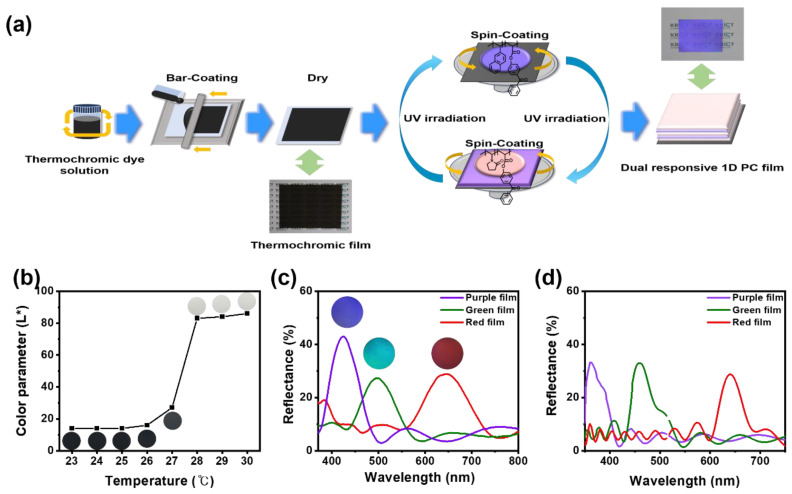
(**a**) Schematic depicting the fabrication method for a thermal and humidity responsive 1D PC multilayer via bar- and spin-coating processes. (**b**) Color parameter (L*) of thermochromic films with temperature. (**c**) Initial reflectance spectra of purple, green, and red films. (**d**) Theoretical reflectance spectra of purple, green, and red films via ellipsometry data.

**Figure 3 polymers-14-05330-f003:**
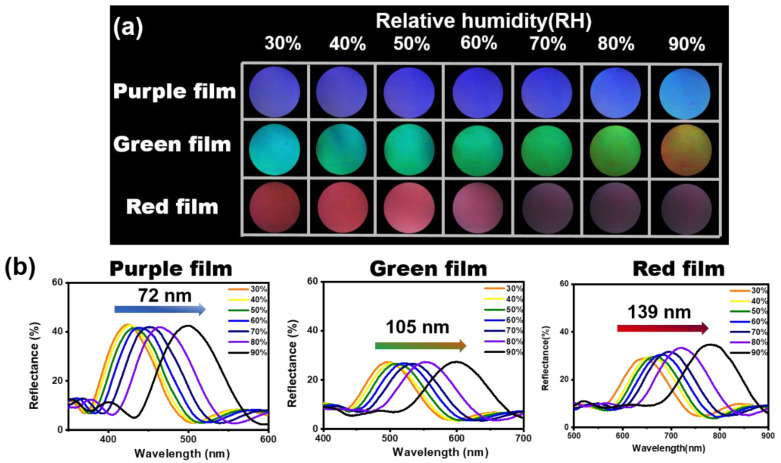
(**a**) Images of three films with black background after showing color changes when exposed to 30 to 98% RH level. (**b**) Reflectance spectra of three films as RH increased from 10 to 98%.

**Figure 4 polymers-14-05330-f004:**
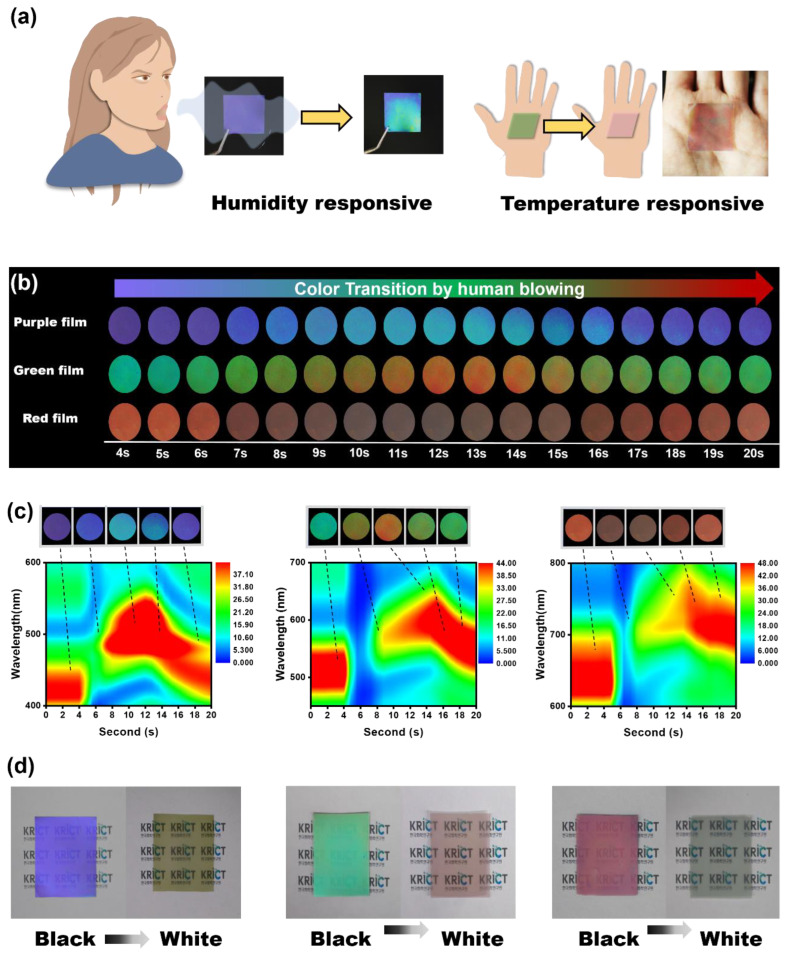
(**a**) Schematic illustration of high-humidity and temperature responsive films. (**b**) Images of three films with a black background after showing color changes caused by human blowing. (**c**) Dynamic reflectance spectra of three films based on human blowing. (**d**) Images of three films depending on black and white backgrounds with temperature.

**Figure 5 polymers-14-05330-f005:**
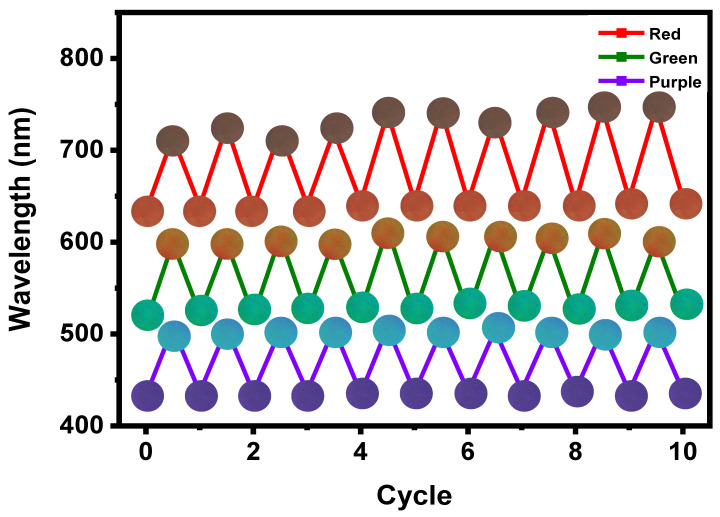
Changes in the maximum PSB after 10 cycles of alternating exposure of three films to human blowing.

**Table 1 polymers-14-05330-t001:** Fabrication conditions and thickness information.

Samples (Films)	Polymer Information		Spin-Coating Speed	Thickness via Ellipsometry (nm)	
	High-Refractive-Index Polymer Solution Concentration	Low-Refractive-Index Polymer Solution Concentration		High-Refractive-Index Polymer	Low-Refractive-Index Polymer
Purple	P(2VN-co-BPA) 1.6%	P(4VP-co-BPA) 1.6%	2500	50	90
Green	P(2VN-co-BPA) 1.6%	P(4VP-co-BPA) 2.0%	2500	50	130
Red	P(2VN-co-BPA) 1.6%	P(4VP-co-BPA) 2.6%	2500	50	165

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
