# Peer review of "Dual Responsive Dependent Background Color Based on Thermochromic 1D Photonic Crystal Multilayer Films"

_polymers, 2022, doi:10.3390/polym14235330_

Round 1

Reviewer 1 Report

Dear Editor

In this work, authors presented dual responsive one-dimensional (1D) photonic crystal (PC) multilayer films that utilize high humidity environment and temperature. 

In my opinion, the paper deserves publication after performing the following a major revision as below:

-  The research gap is not clear. The main challenges and issues from literature should be given in the introduction before giving the objective of your work.

How was thickness of the films determined using ellipsometry?

-          652 mJ/cm2 UV light should be “652 mW/cm2”

-          Authors stated “the thickness of the high and low refractive index polymers by varying the solvent type, concentration of the copolymer in the solution, and spin-coating speed”. A table should be presented in which data on the concentration, spin-speed and thickness change have to be given.

-          Authors stated “it was compared with the theoretical PSB, which was in good agreement with the experimental results.” Where is the validation? I cannot see it.

-          Line 39, ‘Huang’ should be ‘Huang et al.’

-          Authors stated “the thickness of 1D PC films increases with humidity due to the swelling of LP layers, leading the red shift in the PSB”. In my understanding, swelling of the films lead to the decrement in the stacking compact, which in turn acts to increase the band gap and hence blue shifting. Further explanation should be given on that to confirm how red shifting is possible?

-          What are the human blowing properties in terms of humidity and temperature. Information on this is necessary.

Author Response

In this work, authors presented dual responsive one-dimensional (1D) photonic crystal (PC) multilayer films that utilize high humidity environment and temperature.

In my opinion, the paper deserves publication after performing the following a major revision as below:

  1. The research gap is not clear. The main challenges and issues from literature should be given in the introduction before giving the objective of your work.

Response: We appreciate your insightful advice. As pointed out, we added the main challenges and issues of this study at the beginning of Introduction in the revised manuscript (page 1).

The added text is as follows:

“The rapid rise in counterfeit products is raising interest in anti-counterfeiting tech-nology. Consequently, various anti-counterfeiting technologies such as radio frequency iden-tification (RFID), quick response (QR) codes, barcodes, holograms, and color-changing technologies have been developed. Photonic crystals (PCs), one of the color-changing technologies, are attracting attention owing to their color tunability, fast response time, low cost, and remarkable reversibility and repeatability.”

  1. How was thickness of the films determined using ellipsometry?

Response: A single layer of each of P(2VN-co-BPA) and P(4VP-co-BPA) was prepared by surface immobilization onto a silicon wafer and analysis with ellipsometry analysis.

  1. 652 mJ/cm2 UV light should be “652 mW/cm2”

Response: As pointed out by the reviewer, UV lamp light intensity was calculated through conveyor speed (m/Min); thus, 652 mJ/cm2 has correct unit.

  1. Authors stated “the thickness of the high and low refractive index polymers by varying the solvent type, concentration of the copolymer in the solution, and spin-coating speed”. A table should be presented in which data on the concentration, spin-speed and thickness change have to be given.

Response: As pointed out by the reviewer, the thickness of each polymer layers can be controlled by solvent type, concentration of the copolymer in the solution, and spin-coating speed. However, when 1D PC multilayer films were fabricated, we only changed low refractive index polymer solution concentration as mentioned in Experimental. Furthermore, in this revision, we added Table 1 to list thickness and coating information.

Table 1 was added on page 3.

  1. Authors stated “it was compared with the theoretical PSB, which was in good agreement with the experimental results.” Where is the validation? I cannot see it.

Response: Figure 2c and 2d exhibited measured reflectance data and theoretical calculated reflectance data, respectively. The measured PSB is same as theoretical PSB.

  1. Line 39, ‘Huang’ should be ‘Huang et al.’

Response: We have corrected this as recommended.

  1. Authors stated “the thickness of 1D PC films increases with humidity due to the swelling of LP layers, leading the red shift in the PSB”. In my understanding, swelling of the films lead to the decrement in the stacking compact, which in turn acts to increase the band gap and hence blue shifting. Further explanation should be given on that to confirm how red shifting is possible?

Response: PBS shows red-shift with color change because the thickness of low refractive index layers increase with RH values and difference in refractive indices between P(2VN-co-BPA) and P(4VP-co-BPA) layers became larger by humidity. The schematic in the reference paper should be referred to details (Macromolecules 2021, 54, 2, 621-628).

  1. What are the human blowing properties in terms of humidity and temperature. Information on this is necessary.

Response: The photonic crystal film only responds to high humidity (human blowing), and the thermochromic coating layer responds only to human body temperature. The temperature range was set so that it did not change in the temperature of the breath.

Reviewer 2 Report

This paper uses dual responsive 1D photonic crystal multilayer films to utilize a high humidity environment and temperature. high and low refractive index polymers are used to fabricate thermal and high humidity responsive 1D PCs. The title of the article is interesting for the readers. The text of the article is also well-written. A few comments should be considered before publishing.

1. 1D photonic crystals can confine light in only one direction. But 2D photonic crystals can limit light in two directions. Many papers have been reported so far that used 2D photonic crystals for various sensors. The author can also refer to these articles and explain why a one-dimensional photonic crystal is chosen.

Examples:

https://doi.org/10.3390/s130404694,  https://doi.org/10.1016/j.optlastec.2021.107397

https://doi.org/10.1007/s11082-022-03945-9, https://doi.org/10.1166/sl.2013.2726

https://doi.org/10.1166/sl.2015.3517

2. To evaluate the results of this design, these results should be compared with similar articles and important parameters should be compared. In this case, the importance of the output results can be understood.

3. In my opinion, the innovation of this structure should be bolded in the abstract.

Author Response

This paper uses dual responsive 1D photonic crystal multilayer films to utilize a high humidity environment and temperature. High and low refractive index polymers are used to fabricate thermal and high humidity responsive 1D PCs. The title of the article is interesting for the readers. The text of the article is also well-written. A few comments should be considered before publishing.

  1. 1D photonic crystals can confine light in only one direction. But 2D photonic crystals can limit light in two directions. Many papers have been reported so far that used 2D photonic crystals for various sensors. The author can also refer to these articles and explain why a one-dimensional photonic crystal is chosen.

Examples:

https://doi.org/10.3390/s130404694,  https://doi.org/10.1016/j.optlastec.2021.107397

https://doi.org/10.1007/s11082-022-03945-9, https://doi.org/10.1166/sl.2013.2726

https://doi.org/10.1166/sl.2015.3517

Response: As pointed out by the reviewer, we added the following sentence in the revised manuscript (page 2).

“Compared to three-dimensional (3D) and two-dimensional (2D) PCs, one-dimensional (1D) PCs have several advantages, including simple structure, good optical properties, simple manufacturing process, and low cost.”

  1. To evaluate the results of this design, these results should be compared with similar articles and important parameters should be compared. In this case, the importance of the output results can be understood.

Response: This concept is yet to be implemented by anyone, and there is nothing comparable with it. This is because typical colorimetric or electric humidity sensors are intended to measure lower humidity and are not designed to only respond to high humidity (over 90 % RH respond).

  1. In my opinion, the innovation of this structure should be bolded in the abstract.

Response: Thank you for your kind comment. Bold text is not allowed by regulations of Polymers.

Round 2

Reviewer 1 Report

Dear Editor

The authors have considered most of the major comments in the revised version, so it can be accepted now.

Author Response

Reviewer #1:

The authors have considered most of the major comments in the revised version, so it can be accepted now.

Response: Thank you very much for your thorough review.

Reviewer 2 Report

These comments have not yet been properly answered in the revised manuscript:

For comment 1, an explanation has been added to the text, but no references have been added. Please also add references for the sentence added to the text.

#Comment 1. 1D photonic crystals can confine light in only one direction. But 2D photonic crystals can limit light in two directions. Many papers have been reported so far that used 2D photonic crystals for various sensors. The author can also refer to these articles and explain why a one- dimensional photonic crystal is chosen. Examples:

https://doi.org/10.3390/s130404694https://doi.org/10.1016/j.optlastec.2021.107397

https://doi.org/10.1007/s11082-022-03945-9, https://doi.org/10.1166/sl.2013.2726

https://doi.org/10.1166/sl.2015.3517

What I mean by bolding innovation is that the content of innovation in the abstract is clear to the reader. In other words, one sentence can be added to the abstract that properly introduces the innovation of the article.

#Comment 3. In my opinion, the innovation of this structure should be bolded in the abstract.

Author Response

Reviewer #2:

These comments have not yet been properly answered in the revised manuscript:

For comment 1, an explanation has been added to the text, but no references have been added. Please also add references for the sentence added to the text.

#Comment 1. 1D photonic crystals can confine light in only one direction. But 2D photonic crystals can limit light in two directions. Many papers have been reported so far that used 2D photonic crystals for various sensors. The author can also refer to these articles and explain why a one- dimensional photonic crystal is chosen. Examples:

https://doi.org/10.3390/s130404694,  https://doi.org/10.1016/j.optlastec.2021.107397

https://doi.org/10.1007/s11082-022-03945-9, https://doi.org/10.1166/sl.2013.2726

https://doi.org/10.1166/sl.2015.3517

Response: Based on the reviewer's comments, we have added relevant reference.

What I mean by bolding innovation is that the content of innovation in the abstract is clear to the reader. In other words, one sentence can be added to the abstract that properly introduces the innovation of the article.

#Comment 3. In my opinion, the innovation of this structure should be bolded in the abstract.

Response: Response: We appreciate your insightful advice. Following the reviewer's comment, we added a statement about the great promise of our study to the abstract.

The added text is as follows:

“To the best of knowledge, it is the first report of a dual response colorimetric films that changes color in various ways depending on temperature and humidity changes, and we believe that it can be applied to various applications.”